# Hyperbaric Oxygen Therapy as a Novel Approach to Modulating Macrophage Polarization for the Treatment of Glioblastoma

**DOI:** 10.3390/biomedicines12071383

**Published:** 2024-06-21

**Authors:** Chun-Man Yuen, Hung-Pei Tsai, Tzu-Ting Tseng, Yu-Lung Tseng, Ann-Shung Lieu, Aij-Lie Kwan, Alice Y. W. Chang

**Affiliations:** 1Institute of Basic Medical Sciences, National Cheng Kung University, Tainan 701, Taiwan; miconeuron@gmail.com; 2Division of Neurosurgery, Department of Surgery, Kaohsiung Chang Gung Memorial Hospital, Kaohsiung 833, Taiwan; 3School of Medicine, College of Medicine, Chang Gung University, Taoyuan 333, Taiwan; 4Division of Neurosurgery, Department of Surgery, Kaohsiung Medical University Hospital, Kaohsiung Medical University, Kaohsiung 807, Taiwan; carbugino@gmail.com (H.-P.T.); cawaii7992@gmail.com (T.-T.T.); e791125@gmail.com (A.-S.L.); 5Department of Neurology, Kaohsiung Chang Gung Memorial Hospital, College of Medicine, Chang Gung University, Kaohsiung 333, Taiwan; carbu010200@gmail.com; 6Department of Surgery, School of Medicine, College of Medicine, Kaohsiung Medical University, Kaohsiung 807, Taiwan; 7Department of Neurosurgery, University of Virginia, Charlottesville, VA 22904, USA; 8Department of Physiology, College of Medicine, National Cheng Kung University, Tainan 701, Taiwan; 9Cheng-Hsing Campus, College of Medicine, National Cheng Kung University, Tainan 701, Taiwan

**Keywords:** GBM, HBO, macrophage polarization, tumor-associated macrophages, apoptosis

## Abstract

Glioblastoma multiforme (GBM) is a highly aggressive brain cancer with a poor prognosis despite current treatments. This is partially attributed to the immunosuppressive environment facilitated by tumor-associated macrophages, which predominantly underlie the tumor-promoting M2 phenotype. This study investigated the potential of hyperbaric oxygen (HBO) therapy, traditionally used to treat conditions such as decompression sickness, in modulating the macrophage phenotype toward the tumoricidal M1 state and disrupting the supportive tumor microenvironment. HBO has direct antiproliferative effects on tumor cells and reduces hypoxia, which may impair angiogenesis and tumor growth. This offers a novel approach to GBM treatment by targeting the role of the immune system within the tumor microenvironment. The effects of HBO on macrophage polarization and GBM cell viability and apoptosis were evaluated in this study. We detected that HBO promoted M1 macrophage cytokine expression while decreasing GBM cell viability and increasing apoptosis using GBM cell lines and THP-1-derived macrophage-conditioned media. These findings suggest that HBO therapy can shift macrophage polarization toward a tumoricidal M1 state. This can improve GBM cell survival and offers a potential therapeutic strategy. In conclusion, HBO can shift macrophages from a tumor-promoting M2 phenotype to a tumoricidal M1 phenotype in GBM. This can facilitate apoptosis and, in turn, improve treatment outcomes.

## 1. Introduction

Glioblastoma multiforme (GBM) is among the most aggressive and lethal brain cancers in adults. It is characterized by a rapid progression and a propensity for widespread infiltration into surrounding brain tissues. It also represents the worst glial cell tumor and presents challenges in neuro-oncology due to its marked cellular and molecular heterogeneity. Clinically, patients diagnosed with GBM often have a poor prognosis, with a median survival duration of approximately 15 months, despite aggressive therapeutic strategies that typically combine surgical resection, radiation, and chemotherapy [1,2,3]. Macrophages are versatile cells of the innate immune system. They have long been implicated in tumor biology and are capable of both tumoricidal and tumor-promoting actions [4]. With GBM, macrophages are recruited to the tumor microenvironment and are often co-opted. This contributes to its malignancy [5]. GBM is difficult to treat, and the current standard of care only modestly extends patient survival. This therapeutic challenge is exacerbated by the immunosuppressive milieu of the GBM microenvironment, which is partly orchestrated by tumor-associated macrophages (TAMs). An increasing number of reports have shown that TAMs in cancer predominantly assume an M2-like phenotype. This is associated with tissue repair and remodeling, immunosuppression, and the promotion of angiogenesis, which facilitate tumor growth and survival [6,7].

Several studies have investigated the regulation of the function of macrophages for the treatment of GBM. The inherent ability of macrophages to polarize into different functional states, ranging from the classically activated proinflammatory M1 phenotype to the alternatively activated anti-inflammatory M2 phenotype, provides a promising target for therapeutic intervention. The aim is to reeducate or reprogram TAMs from a tumor-supportive M2-like state toward a tumoricidal M1-like state [8,9,10]. This will convert them from promoters to inhibitors of the tumor. Emerging therapies targeting macrophages in GBM are aimed at depleting M2-like TAMs, preventing their recruitment and polarization, or reprogramming them toward an M1-like phenotype [11].

Hyperbaric oxygen (HBO) therapy, a medical treatment that involves the administration of 100% oxygen at elevated atmospheric pressure, has traditionally been used to treat decompression sickness, carbon monoxide poisoning, and chronic wounds [12,13,14]. Recent explorations of its usefulness in treating cancers have revealed its significant effects on the tumor microenvironment [15,16]. Increased oxygen concentrations can increase the production of reactive oxygen species, which can induce cell death or sensitize tumor cells to subsequent therapies, such as chemotherapy or radiation [17,18]. Increased oxygen availability can also improve the efficacy of radiation therapy by enhancing oxygen-dependent radiation damage in tumor cells [19]. In addition to these direct effects, HBO may also modulate angiogenesis within tumors. Hypoxia-inducible factors (HIFs), which are stabilized under low-oxygen conditions, promote the formation of new blood vessels to supply rapidly growing tumors [20,21]. Therefore, the reduction of hypoxia by HBO may inhibit the HIF-mediated angiogenic pathway and disrupt the nutrient and oxygen supplies required for tumor growth. This study aimed to explore the effectiveness of HBO in modulating macrophage polarization for the M1 and M2 phenotypes for the treatment of GBM.

## 2. Materials and Methods

### 2.1. Cell Cultures, Macrophage Polarization, and Reagents

The human glioblastoma cell lines GBM8401 and T98G including red fluorescent protein (RFP), along with the monocyte cell line THP-1, were acquired from the American Type Culture Collection in Manassas, VA, USA. They were maintained in RPMI 1640 medium (Gibco, New York, NY, USA) enriched with 10% fetal bovine serum and 1% penicillin–streptomycin, under a 5% CO_2_ atmosphere at 37 °C. For macrophage polarization, THP-1 cells (5 million) were first activated with 320 nM phorbol 12-myristate 13-acetate (Sigma, St. Louis, MO, USA) and incubated at 37 °C for 24 h to produce an M0 macrophage-conditioned medium (M0-CM). To induce the M1 or M2 phenotype, THP-1 cells underwent 6 h phorbol 12-myristate 13-acetate treatment, followed by the addition of M1- (100 ng/mL Lipopolysaccharide and 20 ng/mL interferon-gamma [IFN-γ]) or M2-polarizing agents (20 ng/mL interleukin [IL]-4 and IL-13) and further incubation at 37 °C for 18 h [22]. The supernatants were collected and designated as M1 macrophage-conditioned media (M1-CM) or M2 macrophage-conditioned media (M2-CM).

### 2.2. Cell Proliferation Assay

Cell viability of GBM8401 and T98G lines post hyperbaric oxygen (HBO) treatment, alone or in combination with macrophage-conditioned media (M0, M1, or M2-CM), was evaluated using the MTT assay. Initially, these cells (30,000 per 0.5 mL per well) were seeded in a 24-well plate and cultured at 37 °C in a 5% CO_2_ environment for 24 h. Subsequently, the cells were treated with M0, M1, or M2-CM and subjected to HBO (100% oxygen at 1.5 atm for 1.5 h every day) for another 24 h before quantification.

### 2.3. Immunohistochemical Staining

Brain tissues were excised for CD86 and CD206 immunohistochemical analysis twenty-one days after the injection of tumor cells. The tissue samples were fixed in formalin, paraffin-embedded, and sectioned at thicknesses of 3 μm. These sections were subsequently deparaffinized, rehydrated, and antigen-retrieved using a Target Retrieval solution (pH 9.0; Dako; S2368; Glostrup, Denmark) in an autoclave at 121 °C for 10 min. Following a 20 min room temperature equilibration, the sections were treated with 3% hydrogen peroxide for 5 min to block endogenous peroxidase activity. They were probed after two rinses with Tris buffer with anti-CD86 and anti-cluster of differentiation 206 (CD206) antibodies for 1 h, washed, and exposed to a horseradish peroxidase-linked secondary antibody for 30 min. After this, 3,3-diaminobenzidine (Dako; K5007; Glostrup, Denmark) was applied for 5 min, followed by counterstaining with Mayer’s hematoxylin for 90 s and mounting with malinol.

### 2.4. Immunohistochemical Staining

The samples were subsequently washed twice with PBS and incubated with mouse anti-CD68 (Invitrogen; 14-0681-82; Waltham, MA, USA) and rabbit anti-CD86 (Proteintech; 26903-1-AP; Chicago, IL, USA) antibodies to mark M1 macrophages, as well as mouse anti-CD68 and rabbit anti-CD206 (Proteintech; 18704-1-AP; Chicago, IL, USA) antibodies to mark M2 macrophages, for 16 h at 4 °C. Following an additional two washes with PBS, the samples were incubated with goat anti-mouse IgG (H+L)-FAM and goat anti-rabbit IgG (H+L)-TAMRA for 90 min at room temperature. Fluorescent images were captured using an Olympus fluorescence microscope (U-RFL-T), and the fluorescence intensity and cell counts were quantified using ImageJ software, version 1.44d (NIH).

### 2.5. Real-Time PCR

The levels of gene expression of tumor necrosis factor-alpha (TNF-α), Interleukin-1β (IL-1β), Interleukin-6 (IL-6), Transforming growth factor-beta (TGF-β), Interleukin-4 (IL-4), and Interleukin-10 (IL-10) were assessed at the functional evaluation endpoint. RNA was extracted using a Total RNA Isolation System and quantified using spectrophotometry. For cDNA synthesis, 1 μg of total RNA was reverse-transcribed in a 20 μL reaction containing 200 units of Superscript II, 0.5 μg of oligo(dT)12-18, 200 pmol of dithiothreitol, 10 pmol of dNTP, and 40 units of ribonuclease inhibitor, according to the Superscript II protocol (Gibco BRL). The cDNA was stored at −20 °C until mRNA analysis. Polymerase chain reaction (PCR) amplification of the target genes was conducted with FastStart Taq DNA polymerase (Roche Diagnostics, Rotkreuz, Switzerland) using 1.5 μL of cDNA per 2 μL reaction, which also included an enzyme buffer, MgCl_2_ (50 pmol), dNTP (200 μmol), and primers (20 pmol). The PCR cycles were as follows: an initial cycle at 94 °C for 1 min, followed by 30 cycles at 60 °C for 30 s, and a final extension at 72 °C for 1 min.

### 2.6. Western Blot Assay

Protein extraction was performed by lysing all samples with 200 μL of lysis buffer, and 50 μg of protein from each sample was electrophoresed on a sodium dodecyl sulfate-polyacrylamide gel at 50 V for 4 h. The proteins were transferred to polyvinylidene fluoride membranes after electrophoresis. The membranes were blocked for 1 h and probed with the following primary antibodies overnight at 4 °C: cleaved caspase-3 antibody (Cell Signaling; #9661; Boston, MA, USA), PARP antibody (Cell Signaling; #9532; Boston, MA, USA), and β-actin (1:20,000, Sigma, A5441, St. Louis, MO, USA). This was followed by a 90 min incubation with secondary antibodies: goat anti-rabbit (1:5000, AP132P; Millipore, Burlington, MA, USA) and goat anti-mouse (1:5000, AP124P; Millipore, Burlington, MA, USA). The proteins were detected using enhanced chemiluminescence (Western Lightning, Waltham, MA, USA; Perkin Elmer, Waltham, MA, USA) and documented using the MiniChemiTM system for image analysis.

### 2.7. Apoptosis Assay

Using the Muse^®^ system (v.1), the cells were treated with Cell Cycle reagent for cell cycle analysis, with outcomes assessed on a Muse^®^ Cell Analyzer. For evaluating apoptosis evaluation, the cells were incubated with 100 μL of Muse^®^ Annexin V and Dead Cell reagent at room temperature mixed with an equal volume of cell suspension (10^5^ cells/mL). Apoptotic cells were analyzed after 20 min using the same system to provide quantitative insights into the cell death and survival mechanisms.

### 2.8. Animal Model

ICR mice, sourced from the LASCO Laboratory Animal Center in Taipei, Taiwan, were maintained in an environment controlled at 24 °C with established light/dark cycles (6:00 am to 6:00 pm) and had unrestricted access to food. For this study, 1 × 10^5^ of C6 cells were stereotactically injected into the striatum of these mice; the volume of the cells was 5 μL. This procedure was replicated in ICR nude mice to evaluate tumor growth during immunocompromise. The mice were divided into two groups: control and HBO. The control group only received an intracranial injection of C6 cells, while the HBO group received an intracranial injection of C6 cells followed by HBO treatment starting one week after the injection. The HBO treatment involved exposure to 100% oxygen at 1.5 atmospheres for 1.5 h daily for 14 days, to assess the therapeutic effects of oxygen under increased pressure on tumor development. The animal protocol received approval from the Kaohsiung Medical University Committee of Institutional Animal Research (IACUC 111223). Each group consisted of 10 mice, a number determined based on statistical requirements to minimize the use of animals. In addition, 0.5 mg/100 g Carprofen was administered, to limit the pain and suffering of the animals to the minimum. Additionally, during injection, treatment, and sampling, we evaluated the animals’ pain and discomfort through behavioral observations and physiological indicators. All animals were euthanized according to an appropriate procedure, including anesthesia, to minimize their suffering.

### 2.9. Statistical Analysis

Statistical analyses were performed using IBM SPSS Statistics (version 24.0; Armonk, NY, USA). The quantitative Western blot data were analyzed using Student’s *t*-test to determine the differences between groups. Survival rates were examined using the Kaplan–Meier survival analysis, which provided insights into time-to-event data. For experiments involving multiple time points or dosages, two-way ANOVA was used to assess interactions and the main effects. Statistical significance was denoted by *p*-values less than 0.05.

## 3. Results

### 3.1. Differential Impact of Macrophage Polarization on Glioblastoma Cell Viability

To explore the impact of macrophage polarization within the tumor microenvironment on the progression and viability of GBM cells, we investigated how conditioned media derived from pro-inflammatory M1 and anti-inflammatory M2 macrophages influenced the survival rates of the GBM8401 and T98G distinct GBM cell lines. In evaluating the effects of macrophage-derived conditioned media on the GBM cell lines GBM8401 and T98G, experimental data revealed an interplay between macrophage polarization and GBM cell viability. When cultured under various macrophage conditions, both GBM cell lines demonstrated differential viability. There was a notable reduction in the viability of GBM8401 cells when exposed to pro-inflammatory M1-CM, with an average viability of approximately 70% across three replicates (Figure 1). This suggested that the inflammatory milieu had a detrimental effect on GBM8401 cell survival. Conversely, anti-inflammatory M2-CM significantly improved GBM8401 viability by approximately to 172%, indicating a tumorigenic environment conducive to GBM survival or proliferation (Figure 1). A similar pattern was observed for the T98G cell line, with M1-CM reducing cell viability to an average of approximately 69% (Figure 1). This reflected the inflammatory conditions that led to the suppression of GBM cell viability. In contrast, M2-CM treatment led to an increase in average viability of approximately to 158% (Figure 1), suggesting that anti-inflammatory conditions may be advantageous for T98G cell endurance or growth.

### 3.2. Macrophage Polarization Modulates Apoptosis in Glioblastoma Cells

To investigate the role of macrophage polarization in the induction of apoptosis in GBM cells, we examined the pro-apoptotic and anti-apoptotic effects of macrophage-derived conditioned media on the GBM8401 and T98G GBM cell lines. The experimental results on the apoptosis rates of GBM cell lines co-cultured with THP-1 macrophages in various conditioned media indicated the influence of macrophage-derived signals on cancer cell apoptosis. For GBM8401 cells, the control condition showed minimal apoptosis, averaging approximately 3.65% (Figure 2). The introduction of M0-CM, which represented a neutral macrophage state, did not lead to a statistically significant change in apoptosis, and an average rate close to 6.65% was maintained (Figure 2). An increase in the apoptosis rate was observed with M1-CM, indicating a robust pro-apoptotic effect of the pro-inflammatory state, with an average apoptosis rate of approximately 23.7% (Figure 2). In contrast, the M2-CM condition, which simulates an anti-inflammatory macrophage environment, resulted in an apoptosis rate slightly higher than that of the control but considerably lower than that of the M1 condition; the average was approximately 5.85% (Figure 2). The T98G cells demonstrated a slightly higher baseline apoptosis rate than the control group at an average of approximately 4.83% (Figure 2). Exposure to M0-CM caused a small increase in the apoptosis rate to an average of approximately 6.63% (Figure 2). M1-CM significantly promoted apoptosis at an average rate of approximately 15.05% (Figure 2). Anti-inflammatory M2-CM treatment yielded an average apoptosis rate of 7.25% (Figure 2), which was higher than that of the control, indicating a lower protective effect than that of M2-CM on GBM8401 cells. To substantiate the role of macrophage polarization in inducing apoptosis in GBM cells, Western blot analysis was used to detect markers of apoptosis, which were specifically cleaved Poly (ADP-ribose) polymerase (PARP), across various experimental groups (Figure 3). These groups included the control, +M0-CM, +M1-CM, and +M2-CM, with assays for the GBM8401 and T98G GBM cell lines. Western blotting results revealed a distinctive expression of cleaved PARP in the +M1-CM group, indicating active apoptosis, which was consistent with the pro-apoptotic effect observed in M1-CM treated cells during the apoptosis assays. In contrast, the control, +M0-CM, and +M2-CM groups did not demonstrate significant concentrations of these apoptotic markers in the GBM8401 or T98G cell lines. This differential expression pattern underscores the specific pro-apoptotic influence of the pro-inflammatory M1 macrophage environment on GBM cells and further confirms the apoptosis-inducing capacity of M1 macrophage polarization in GBM.

### 3.3. Hyperbaric Oxygen Treatment Induces Macrophage Polarization towards Tumoricidal M1 Phenotype in Glioblastoma Co-Culture Systems

To demonstrate the effect of hyperbaric oxygen (HBO) on macrophage polarization, we utilized immunofluorescence staining to detect M1/M2 biomarkers. THP-1 cells were co-cultured with either GBM8401 or A172 glioblastoma cell lines and subjected to HBO treatment. Immunofluorescence staining was performed to detect the expression of M1 marker CD86 and M2 marker CD206. The results showed that in the control group, THP-1 cells co-cultured with glioblastoma cells (both GBM8401 and A172 cell lines) exhibited a higher expression of the M2 marker CD206, indicating a predominant M2 macrophage phenotype which is associated with a tumor-promoting environment (Figure 4). The high expression of CD206 in these co-cultures suggests that the presence of glioblastoma cells may contribute to the polarization of THP-1 cells towards an M2 phenotype, which can facilitate tumor growth and immune evasion. However, after the application of hyperbaric oxygen (HBO) treatment, there was a notable shift in the macrophage polarization. Specifically, there was a significant decrease in the expression of the M2 marker CD206 and a concomitant increase in the expression of the M1 marker CD86 (Figure 4). The increase in CD86 expression indicates a shift towards the M1 macrophage phenotype, which is known for its tumoricidal activity and ability to stimulate anti-tumor immune responses. This transition from the M2 to the M1 phenotype suggests that HBO treatment can reprogram the macrophages from a state that supports tumor progression to a state that actively combats the tumor. Quantitative analysis of the immunofluorescence data further confirmed this phenotypic switch. The percentage of M2 macrophages (CD206 positive) was significantly reduced in the HBO-treated groups compared to the control groups, indicating that HBO treatment effectively diminishes the population of tumor-promoting macrophages (Figure 4). Concurrently, there was a significant increase in the percentage of M1 macrophages (CD86 positive) following HBO treatment, highlighting the enhanced presence of tumoricidal macrophages. These findings suggest that HBO therapy not only affects macrophage polarization at the molecular level but also induces a functional shift in the macrophage population within the tumor microenvironment. By reducing the proportion of M2 macrophages and increasing the proportion of M1 macrophages, HBO treatment potentially alters the tumor microenvironment to become more hostile to tumor growth and more conducive to anti-tumor immunity. This shift in macrophage polarization might be a crucial mechanism by which HBO exerts its therapeutic effects in glioblastoma, providing a rationale for its use as an adjunctive therapy in cancer treatment.

### 3.4. HBO Induces Dual Modulation of Pro- and Anti-Inflammatory Cytokine mRNA Expression in Macrophages Co-Cultured with GBM Cells

M1 macrophages are known to release pro-inflammatory cytokines such as TNF-α, IL-1β, and IFN-γ, while M2 macrophages predominantly release anti-inflammatory cytokines including IL-4, IL-10, and TGF-β. THP-1 cells and GBM cells were co-cultured in a 1:1 ratio in 6-well plates. The co-cultures were maintained in standard culture conditions and subjected to HBO treatment. HBO treatment involved exposing the cells to 100% oxygen at 1.5 atmospheres for 1.5 h daily for three consecutive days. After the treatment period, the cells were harvested, and the expression levels of TNF-α, IL-1β, IFN-γ, IL-4, IL-10, and TGF-β were measured using real-time PCR (Figure 5). In macrophages co-cultured with GBM8401 cells, there was a pronounced increase in the mRNA concentrations of M1-associated cytokines following HBO treatment, suggesting that HBO may potentiate the pro-inflammatory response. The expression of TNF-α was markedly elevated after HBO, reaching up to 12.8-fold the level observed after the control treatment (Figure 5). Similarly, the concentrations of IL-1β and IFN-γ markedly increased, indicating an enhanced M1 activation in the presence of HBO (Figure 5). On examining the anti-inflammatory markers, IL-10, IL-4, and TGF-β1 showed significantly increased expressions in the presence of GBM8401 cells. The most remarkable change was observed in the concentration of IL-10, which increased to approximately 12.8 times that of HBO (Figure 5). This suggests that HBO enhances pro-inflammatory signals, and this coincides with the upregulation of anti-inflammatory cytokines. In contrast, macrophages co-cultured with T98G cells showed a different pattern. The mRNA concentrations of M1 cytokines such as TNF-α and IL-1β are notably upregulated, especially with the addition of HBO, indicating a similar trend of M1 activation (Figure 5). However, the expression of anti-inflammatory markers is also substantially increased, especially TGF-β1, which shows an approximate 18.8-fold increase with HBO treatment (Figure 5). These data suggest that HBO, when applied to a microenvironment with GBM cells, can lead to dual modulation of the macrophage phenotype and significantly enhance both pro-inflammatory and anti-inflammatory cytokine expressions. The co-culture of tumor and immune cells tended to lean toward M2 polarization; however, the polarization shifted toward M1 with the addition of HBO.

### 3.5. Modulation of Macrophage Polarization by Hyperbaric Oxygen Therapy in a Mouse Glioma Model

CD86 is a marker of M1 macrophages, which are known for their pro-inflammatory and tumoricidal activities. CD206, on the other hand, is a marker for M2 macrophages, which are generally associated with anti-inflammatory responses and can promote tumor growth. To demonstrate the effectiveness of HBO therapy in inducing the polarization of macrophages toward an M1 phenotype in vivo, which is favorable for anti-tumor immunity, C6 glioma cells were injected into the striatum of mice to establish a brain tumor model.

We quantified the number of CD86 (M1 marker)- and CD206 (M2 marker)-positive cells in glioblastoma (GBM) tissue samples. Immunohistochemically stained sections of the GBM models were analyzed to determine the number of CD86 and CD206-positive cells (Figure 6). The quantitative analysis revealed a significant increase in the number of CD86-positive cells in the HBO-treated group compared to the control group (145.33 ± 15.67 vs. 39.67 ± 5.77, respectively, *p* < 0.01) (Figure 6). This suggests a marked increase in the M1 macrophage population, which is associated with pro-inflammatory and anti-tumorigenic functions. Conversely, the number of CD206-positive cells was significantly reduced in the HBO-treated group compared to the control group (54.33 ± 7.23 vs. 164.67 ± 12.45, respectively, *p* < 0.05) (Figure 6), indicating a decrease in the M2 macrophage population, which is typically linked to anti-inflammatory and pro-tumorigenic activities. These findings suggest that HBO therapy effectively promotes the polarization of macrophages from the tumor-supportive M2 phenotype to the tumoricidal M1 phenotype in the glioblastoma model. This shift in macrophage polarization is favorable for enhancing anti-tumor immunity and potentially inhibiting tumor growth.

### 3.6. Synergistic Enhancement of Apoptosis in Glioblastoma Cells by THP-1 Macrophages and Hyperbaric Oxygen Therapy

To ascertain the interactive effects of HBO and macrophage activity on the apoptotic response of GBM cells, we designed an experiment to evaluate the extent of apoptosis in GBM cell lines subjected to HBO, THP-1 macrophages, or their combination. The co-cultures were subjected to the same HBO treatment (100% oxygen at 1.5 atmospheres for 1.5 h daily for three consecutive days). Following the treatment, the GBM cells were sorted using flow cytometry based on the RFP marker. The sorted GBM cells were then subjected to an apoptosis assay to assess the extent of apoptosis. These results reflect the extents of apoptosis in GBM8401 and T98G GBM cell lines under the influence of HBO, THP-1 macrophages, or their combination and suggest the intricate dynamics of the treatment response. For GBM8401 cells, the baseline apoptosis was modest, with control averages of approximately 4% (Figure 7). HBO did not significantly change this rate, indicating no statistically significant proapoptotic effect. However, the introduction of THP-1 macrophages resulted in a substantial increase in apoptosis, averaging at approximately 16% (Figure 7). The combined treatment of THP-1 macrophages with HBO markedly amplified this effect, increasing the average apoptosis rate to approximately 26.38% (Figure 7). For the T98G cells, the control conditions yielded a similar baseline for apoptosis, averaging just over 4.8% (Figure 7). HBO alone led to a limited increase in apoptosis, which was comparable to the effect observed in GBM8401 cells. THP-1 macrophages alone also led to increased apoptosis, with an average of approximately 13.65% (Figure 7). The apoptosis rates increased when they were combined with HBO, averaging approximately 30.67% (Figure 7). These observations imply a synergistic effect of HBO and macrophage-induced apoptosis in GBM cells, with the combination treatment resulting in the highest levels of cell death in both cell lines. Following the assessment of the apoptotic response in GBM8401 and T98G cell lines after various treatments, we performed Western blot analysis to detect cleaved PARP, which is a hallmark of apoptosis (Figure 8). Western blot analysis to detect cleaved PARP indicated that the groups treated with HBO alone showed no significant increase in apoptotic markers, which was similar for the control groups. Minimal apoptosis was observed in the THP-1-only treated groups, as evidenced by a slight increase in the expression of cleaved PARP. However, a significant increase in the expression of cleaved PARP was observed in groups co-treated with THP-1 macrophages and HBO, indicating a marked apoptotic response. This pattern suggests that HBO alone does not induce apoptosis in GBM cells but its combination with THP-1 macrophages significantly enhances the apoptosis, as reflected by the increased concentrations of apoptosis-specific markers. This synergy reflects the potential of combined therapeutic strategies for enhancing the efficacy of GBM treatment.

### 3.7. Interplay between Hyperbaric Oxygen Therapy and Macrophage Interaction in Modulating Glioblastoma Cell Viability

To investigate the potential synergistic or antagonistic effects of hyperbaric oxygen (HBO) combined with macrophage-mediated immune responses on the survival of GBM cells, we evaluated cell viability following HBO and macrophage interaction in vitro (Figure 9). The present data depict the impact of HBO therapy and THP-1 macrophage co-culture on the viability of two glioblastoma cell lines, GBM8401 and T98G, with measurements taken under various conditions. In GBM8401 cells, the control condition showed robust cell viability with an average hovering rate of approximately 100% (Figure 9). The application of HBO alone caused a negligible decrease in viability, maintaining averages near the baseline. The presence of THP-1 macrophages alone appeared to enhance cell viability, with an average value exceeding that of the control. However, in combination, THP-1 macrophages and HBO therapy led to a marked reduction in cell viability, with averages dropping to approximately 78.97% (Figure 9). This suggested a synergistic effect that compromises GBM8401 cell survival. The T98G cell line, under control conditions, demonstrates a viability close to the baseline, with an average of approximately 100%. Similar to GBM8401, HBO alone had minimal impact on T98G cell viability. Co-culture with THP-1 macrophages alone resulted in increased viability, with averages increasing by approximately 112.17% (Figure 9). However, the combination of THP-1 macrophages and HBO substantially reduced the viability of T98G cells by an average of approximately 76.44% (Figure 9). These results underscore the complex interactions between therapeutic interventions and the tumor microenvironment. They highlight that the combination of HBO with immune cells can significantly alter cell survival although it may have limited direct toxic effects on GBM cells. This provides a potential avenue for enhancing the efficacy of GBM treatments.

## 4. Discussion

The impact of M1 and M2 macrophages on GBM is a nuanced aspect of the tumor microenvironment, which plays a crucial role in the progression or inhibition of tumor development and progression [23]. M1 macrophages are typically associated with a pro-inflammatory response, characterized by the secretion of cytokines such as TNFα, IL-1β, and IFN-γ [24,25]. These cytokines and chemokines are involved in the natural defense mechanisms of the body that initiate a robust immune response [26]. M1 macrophages, with their tumoricidal activity, promote anti-tumor immunity, possibly by inhibiting GBM progression through pro-inflammatory mediators [27,28]. Conversely, M2 macrophages are generally associated with an anti-inflammatory response that supports tumor progression and suppression of the immune defenses of the body against the tumor [29,30]. M2 macrophages express markers such as CD206 and secrete various cytokines and growth factors including TGF-β1, IL-4, and IL-10 [31,32]. These factors contribute to the immunosuppressive environment and are implicated in various pro-tumorigenic processes, including angiogenesis, tumor cell proliferation, and inhibition of cytotoxic T-cell activity [33,34]. In GBM, M2-like macrophages within the tumor microenvironment can promote tumor development by creating conditions that facilitate cancer cell survival, such as the suppression of effective immune surveillance and promotion of tissue remodeling conducive to tumor invasion [35,36]. Furthermore, M2 macrophages may assist in maintaining glioblastoma stem cell populations, contributing to therapy resistance and tumor recurrence [36]. The complex interplay of these two macrophage phenotypes within the glioblastoma microenvironment underscores the dynamic nature of tumor-immune interactions and the potential for therapeutic strategies targeting macrophage polarization. By manipulating the balance toward the M1 phenotype or suppressing the M2 phenotype, it may be possible to enhance the ability of the immune system to combat GBM and improve patient outcomes.

In our previous study, we examined the effect of HBO therapy on GBM cells using both in vitro and in vivo approaches. Initially, the investigation showed that HBO did not significantly affect the proliferation of GBM cells. However, HBO enhanced treatment effectiveness when combined with chemotherapy and radiotherapy, leading to reduced cell viability in vitro. This study focused on the influence of HBO on cancer stem cells (CSCs), a subpopulation of tumors known for their role in tumor initiation, proliferation, and resistance to treatment. This study found that HBO reduced the ability of GBM cells to form CSCs, as evidenced by decreased protein levels of stemness markers such as CD133, OCT4, and SOX2. This suggested an inhibitory effect on CSC formation and maintenance. Further in vivo experiments corroborated these findings, showing a significant reduction in the number of CD133-positive cells and lower concentrations of stemness markers in the treated groups. This indicates the potential of HBO in suppressing CSC formation and maintenance in living organisms and highlights its value as an adjunctive therapy for GBM. This research also delved into the mechanism behind these effects, suggesting that HBO therapy could counter the hypoxic conditions favoring CSC survival and self-renewal by increasing oxygen supply. This, in turn, reduced the expression of HIF-1α and attenuated GBM stemness [19].

In the immune system, macrophages play versatile roles and are often dichotomized into M1 and M2 phenotypes, each characterized by distinct biomarkers and cytokine profiles. M1 macrophages, commonly referred to as ‘classically activated’, are integral to the first line of defense of the body against pathogens and are involved in the initiation of pro-inflammatory responses. In contrast, M2 macrophages, known as ‘alternatively activated’, are associated with tissue repair, resolution of inflammation, and support of tumor growth in the context of cancer [37,38]. Beyond the commonly acknowledged surface markers, CD86 for M1 and CD206 for M2 macrophages, a suite of cytokines offers a broader palette for distinguishing between these phenotypes [39]. TNF-α and IL-1β are important pro-inflammatory cytokines produced predominantly by M1 macrophages. These cytokines serve as effector molecules for destroying pathogens and play critical roles in orchestrating the inflammatory response, promoting the activation of immune cells, and perpetuating the inflammatory cascade [40]. IL-6, a cytokine secreted by M1 macrophages, contributes to the inflammatory milieu and is implicated in the pathogenesis of various inflammatory and autoimmune diseases [41]. In stark contrast, TGF-β, IL-4, and IL-10 are more commonly associated with the M2 phenotype. TGF-β is a potent immunoregulatory cytokine that plays a key role in tissue regeneration and fibrosis, as well as the suppression of immune responses [42]. This contributes to the immunosuppressive environment favored by tumors. IL-4 and IL-10 are anti-inflammatory cytokines that facilitate the differentiation of macrophages toward the M2 phenotype and mediate the suppression of pro-inflammatory responses to facilitate tissue repair and remodeling [25]. The expressions of these cytokines provide a functional readout of the macrophage state: TNF-α, IL-1β, and IL-6 represent a pro-inflammatory, anti-tumoral M1 phenotype, while TGF-β, IL-4, and IL-10 represent an anti-inflammatory, pro-tumoral M2 phenotype.

In this study, HBO therapy was explored as a potential modulator of the GBM microenvironment, with a focus on its influence on macrophage polarization and the subsequent effects on tumor progression. HBO can induce dual modulation of the expressions of pro- and anti-inflammatory cytokine mRNAs when macrophages are co-cultured with GBM cells. This dual response is characterized by an increase in pro-inflammatory cytokines such as TNF-α, IL-1β, and IFN-γ, which are associated with M1 macrophage activation, along with increased expressions of anti-inflammatory cytokines such as IL-4, IL-10, and TGF-β1, indicative of M2 macrophage activity. The interaction between HBO and macrophage activity has been demonstrated to significantly influence apoptosis in GBM cells. When THP-1 macrophages and HBO were combined, a synergistic effect was observed, leading to a marked increase in the apoptotic rate of GBM cells. This was further corroborated by cleaved PARP, a marker of apoptosis. This suggests that HBO in combination with macrophages potentiates the apoptotic response, although it may not induce significant apoptosis alone. In terms of cell viability, the combination of HBO and macrophage interactions yielded complex effects on GBM cell lines. HBO alone did not significantly affect GBM cell viability, but the introduction of THP-1 macrophages appeared to enhance cell viability. When both were applied together, a notable decrease in cell viability was observed, suggesting that the interaction between HBO and macrophage activity can significantly compromise GBM cell survival and enhance the efficacy of GBM treatment.

## 5. Conclusions

In conclusion, this study presents an extensive exploration of the therapeutic potential of HBO in altering the tumor microenvironment of GBM through macrophage polarization. Studies have highlighted that HBO has a marked effect on macrophage phenotype, shifting the balance from the tumor-promoting M2 state toward the tumoricidal M1 state, as indicated by increased expression of CD86 and decreased expression of CD206. This shift in polarization is associated with enhanced pro-inflammatory and anti-tumorigenic responses within the tumor microenvironment. Furthermore, the influence of macrophage-derived conditioned media on GBM cell viability and apoptosis elucidated the complex interplay between macrophage phenotype and GBM cell survival. The detrimental effect of pro-inflammatory M1 macrophages on GBM cell viability, as well as their role in promoting apoptosis, contrasts with the supportive effects of the M2 phenotype. These findings underscore the dual modulatory role of HBO in cytokine expression, revealing the simultaneous upregulation of pro- and anti-inflammatory cytokines in macrophages co-cultured with GBM cells. This suggests a nuanced immune regulatory role for HBO that may extend beyond simply enhancing proinflammatory responses. Additionally, the synergistic effect of HBO and macrophage interactions in inducing apoptosis in GBM cells provides a promising outlook for combined treatment strategies. This synergy can lead to a significant reduction in cell viability. Therefore, HBO can potentiate anti-tumor immunity by modulating macrophage polarization and enhancing the apoptotic response of GBM cells. This treatment approach can open new avenues for the development of adjunctive therapies for the management of glioblastoma and improve patient outcomes by targeting complex cellular interactions within the tumor microenvironment.

## Figures and Tables

**Figure 1 biomedicines-12-01383-f001:**
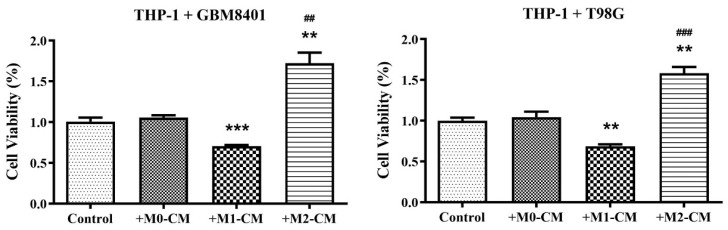
Impact of inflammatory conditions on GBM cell viability. Cell viability was measured using an MTT assay to determine the effects of inflammatory conditions on GBM cells. THP-1 cells co-cultured with GBM8401 or T98G cells were exposed to conditioned media from macrophages polarized to M0, M1, and M2 states. The viability of GBM cells was assessed following exposure. The control group included GBM cells in co-culture without conditioned media. Data represent mean ± SEM of three independent experiments. ** *p* < 0.01, *** *p* < 0.001 compared to control; ^##^
*p* < 0.01, ^###^
*p* < 0.001 on comparing M1-CM and M2-CM to M0-CM. THP-1, a human monocytic cell line; GBM, glioblastoma multiforme; CM, conditioned media; M0, M1, and M2, macrophage polarization states. Control: Cells without conditioned medium. +M0-CM: Cells with conditioned medium from M0 macrophages. +M1-CM: Cells with conditioned medium from M1 macrophages. +M2-CM: Cells with conditioned medium from M2 macrophages.

**Figure 2 biomedicines-12-01383-f002:**
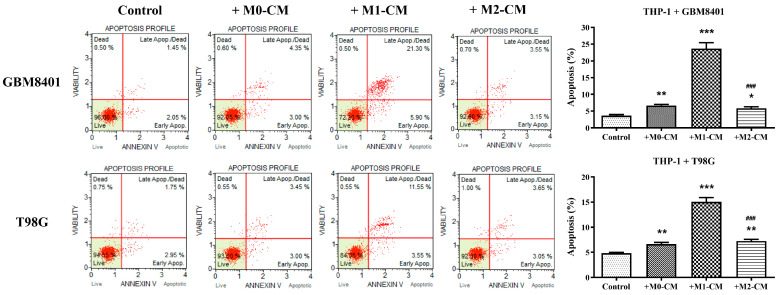
Apoptotic response of GBM cells to macrophage-conditioned media. Flow cytometry analysis was performed to assess apoptosis in GBM cells co-cultured with THP-1 cells under different conditions. The top and bottom panels show GBM8401 and T98G cells, respectively. Each panel shows the apoptosis profile under control conditions and after treatment with conditioned media from M0, M1, and M2 polarized macrophages (+M0-CM, +M1-CM, +M2-CM). The right graphs summarize the percentage of apoptotic cells for both cell lines in each condition. Data represent mean ± SEM from three replicates. * *p* < 0.05, ** *p* < 0.01, *** *p* < 0.001 indicate significance relative to the control; ^###^
*p* < 0.001 indicate significance of M1-CM and M2-CM treatment effects relative to those of M0-CM. Control: Cells without conditioned medium. +M0-CM: Cells with conditioned medium from M0 macrophages. +M1-CM: Cells with conditioned medium from M1 macrophages. +M2-CM: Cells with conditioned medium from M2 macrophages.

**Figure 3 biomedicines-12-01383-f003:**
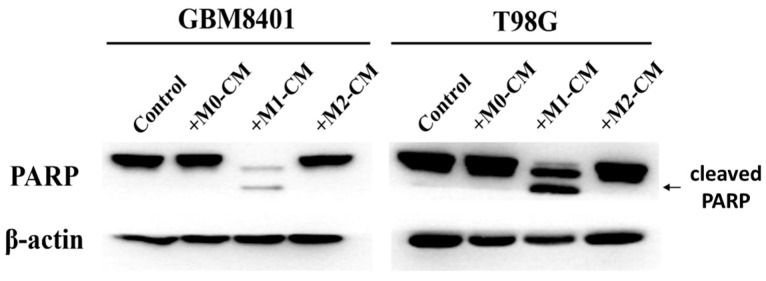
Western blot analysis of PARP in GBM cells. Protein extracts from GBM8401 and T98G cells treated with conditioned media from unstimulated (M0) and polarized (M1 and M2) macrophages were subjected to Western blotting to detect the concentrations of PARP. β-actin was used as a loading control. Control: Cells without conditioned medium. +M0-CM: Cells with conditioned medium from M0 macrophages. +M1-CM: Cells with conditioned medium from M1 macrophages. +M2-CM: Cells with conditioned medium from M2 macrophages.

**Figure 4 biomedicines-12-01383-f004:**
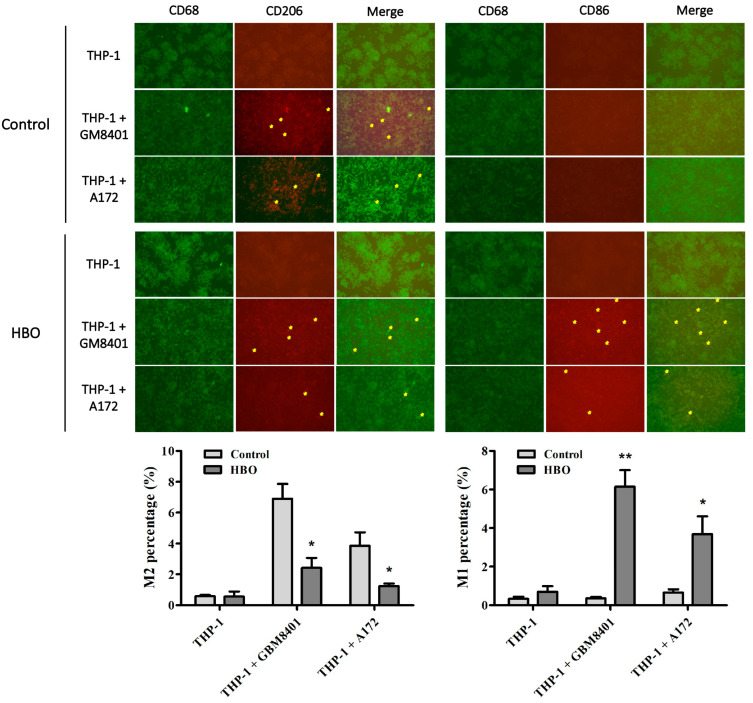
Immunofluorescence staining for macrophage polarization markers CD86 and CD206 in THP-1 cells co-cultured with glioblastoma cells under control and HBO treatment conditions. Representative images of immunofluorescence staining for CD68 (macrophage marker), CD206 (M2 marker), and CD86 (M1 marker). THP-1 cells were co-cultured with GBM8401 or A172 cells. The merged images show the colocalization of the markers, with yellow arrows indicating positive staining. Control Groups: THP-1 cells without HBO treatment. HBO Groups: THP-1 cells with HBO treatment. Quantitative analysis of the percentage of M2 and M1 macrophages. The left graph shows the percentage of M2 macrophages (CD206 positive) in control and HBO-treated groups, while the right graph displays the percentage of M1 macrophages (CD86 positive) in control and HBO-treated groups. Data are presented as mean ± SD. * *p* < 0.05, ** *p* < 0.01 compared to the control group. Control Groups: THP-1 cells without HBO treatment. HBO Groups: THP-1 cells with HBO treatment.

**Figure 5 biomedicines-12-01383-f005:**
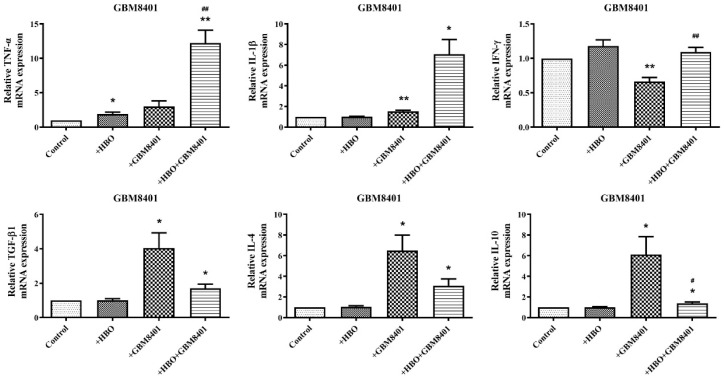
Gene expression analysis in GBM cells following treatment. Quantitative PCR was used to assess the levels of expression of inflammatory and apoptotic markers in GBM8401 and T98G cells. The top panels show the relative mRNA concentrations of TNFα, IL-1β, IFN-γ, TGF-β1, IL-4, and IL-10 in GBM8401 cells. The bottom panels show the expression levels of the same genes in T98G cells. The treatments included control conditions, hyperbaric oxygen (HBO) therapy alone, co-culture with THP-1 cells (+THP-1), and co-culture with HBO therapy (+HBO+THP-1). Gene expression levels are normalized to a housekeeping gene and represented as fold change relative to the control. Data are expressed as mean ± SEM of triplicate experiments. * *p* < 0.05, ** *p* < 0.01, *** *p* < 0.001 compared with the control; ^#^
*p* < 0.05, ^##^
*p* < 0.01 indicate significant differences between treatment groups. Control: THP-1 cells without any treatment. +HBO: THP-1 cells treated with hyperbaric oxygen alone. +GBM8401: THP-1 cells co-cultured with GBM8401 cells. +HBO+GBM8401: THP-1 cells treated with both HBO and co-cultured with GBM8401 cells. +T98G: THP-1 cells co-cultured with T98G cells. +HBO+T98G: THP-1 cells treated with both HBO and co-cultured with T98G cells.

**Figure 6 biomedicines-12-01383-f006:**
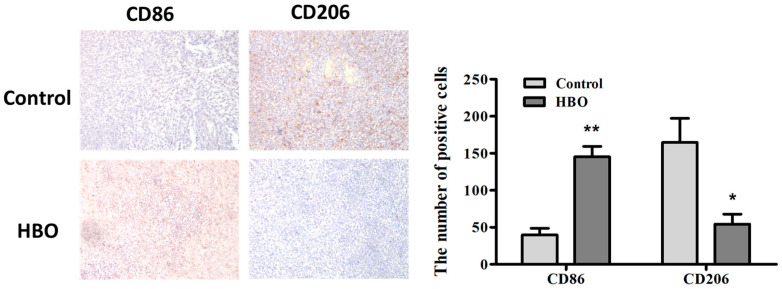
Histological examination of tissue samples after HBO treatment in a GBM animal model. Representative immunohistochemically (IHC) stained sections of glioblastoma (GBM) models are presented at 100× magnification. The top panels show sections stained for CD86, a marker for M1 macrophages. The left image represents the control group, displaying low baseline expression of CD86. The right image represents the HBO-treated group, showing a significant increase in CD86 expression, indicating a higher presence of M1 macrophages. The bottom panels show sections stained for CD206, a marker for M2 macrophages. The left image represents the control group, showing higher expression of CD206, indicative of a predominance of M2 macrophages. The right image represents the HBO-treated group, demonstrating a significant decrease in CD206 expression, suggesting a reduction in M2 macrophages and a shift towards M1 macrophage polarization. The quantitative analysis of the number of CD86- and CD206-positive cells in both control and HBO-treated groups. The data are presented as mean ± SD. * *p* < 0.05, ** *p* < 0.01 compared to the control group. Control: C6 glioma injection without any treatment. HBO: C6 glioma injection with HBO treatment.

**Figure 7 biomedicines-12-01383-f007:**
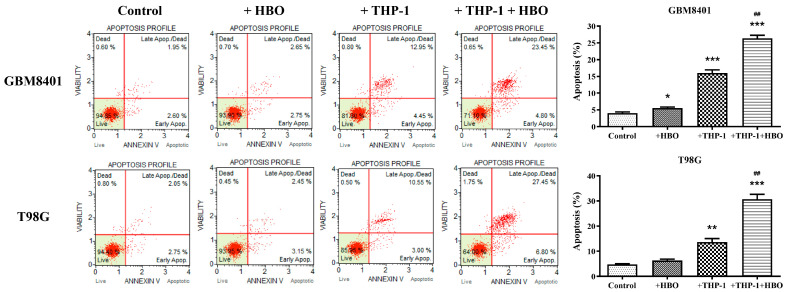
Apoptotic effects of HBO and THP-1 co-culture on GBM cell lines. Apoptosis was evaluated in GBM8401 and T98G cells through flow cytometry, as shown in the left panels. The percentage of apoptotic cells was quantified, as shown in the bar graphs on the right. Cells were treated under control conditions, with hyperbaric oxygen (HBO) therapy, in co-culture with THP-1 cells, and with a combination of THP-1 co-culture and HBO therapy (+THP-1+HBO). The flow cytometry plots categorize cells as live, early-apoptotic, late-apoptotic/dead, and dead, while the bar graphs summarize the percentages of apoptotic cells for each treatment condition for GBM8401 and T98G cell lines. Data are presented as mean ± SEM. Statistical significance is indicated as * *p* < 0.05, ** *p* < 0.01, *** *p* < 0.001 compared to control, and ^##^
*p* < 0.01 for the comparison of treatment groups. Control: Glioblastoma cells without any treatment. +HBO: Glioblastoma cells treated with HBO alone. +THP-1: Glioblastoma cells co-cultured with THP-1 macrophages. +THP-1+HBO: Glioblastoma cells co-cultured with THP-1 macrophages and treated with HBO.

**Figure 8 biomedicines-12-01383-f008:**
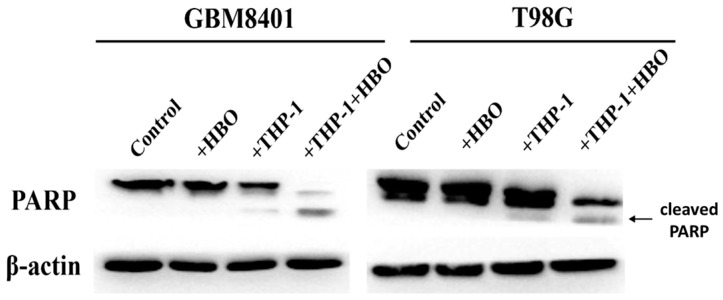
Protein analysis of apoptotic markers in GBM cell lines under various treatments. Western blot assays were conducted to detect PARP cleavage in GBM8401 and T98G cells. The cells were subjected to control conditions, hyperbaric oxygen (HBO) therapy, co-culture with THP-1 cells, and a combination of THP-1 co-culture and HBO therapy. The left panel shows the results for GBM8401 cells, while the right panel shows the results for T98G cells. PARP cleavage is indicative of apoptosis. β-actin was probed as a loading control to ensure equal protein loading across all conditions. Control: Glioblastoma cells without any treatment. +HBO: Glioblastoma cells treated with HBO alone. +THP-1: Glioblastoma cells co-cultured with THP-1 macrophages. +THP-1+HBO: Glioblastoma cells co-cultured with THP-1 macrophages and treated with HBO.

**Figure 9 biomedicines-12-01383-f009:**
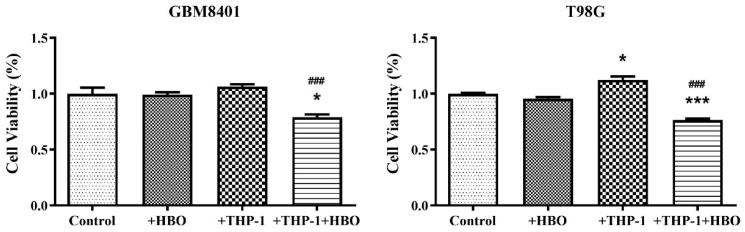
Cell viability of GBM cell lines under different treatment conditions. The bar graphs represent the relative cell viability of GBM8401 (left) and T98G (right) cell lines after treatment with hyperbaric oxygen (HBO) therapy, co-culture with THP-1 cells, and a combination of THP-1 co-culture and HBO therapy relative to the control untreated conditions. Viability was quantified using a standard MTT assay, and the results are normalized to the control group. Data are expressed as mean ± SEM of three independent experiments. Asterisks indicate statistically significant differences in the effects of the treatments and control (* *p* < 0.05, *** *p* < 0.001), while hashes indicate significant differences in the effects of other treatments relative to those of +THP-1 (^###^
*p* < 0.001). Control: Glioblastoma cells without any treatment. +HBO: Glioblastoma cells treated with HBO alone. +THP-1: Glioblastoma cells co-cultured with THP-1 macrophages. +THP-1+HBO: Glioblastoma cells co-cultured with THP-1 macrophages and treated with HBO.

## Data Availability

Data are contained within the article.

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
