# Peer review of "Hyperbaric Oxygen Therapy as a Novel Approach to Modulating Macrophage Polarization for the Treatment of Glioblastoma"

_biomedicines, 2024, doi:10.3390/biomedicines12071383_

Round 1

Reviewer 1 Report

Comments and Suggestions for Authors

The manuscript intends to demonstrate application of HBO as a method to modulate macrophage polarization to enhance apoptosis in glioblastoma in in vitro and in vivo model. The attempt of the authors has been descent and slightly novel, although several previous reports of HBO as an alternative adjuvant therapy in cancer treatment are available. The authors have made a descent attempt to demonstrate effect of HBO in presence of M1 polarized macrophage, however, there is no conclusive evidence/experiment designed to demonstrate that HBO triggers M2àM1 polarization as claimed in the title of the manuscript. The authors may attempt to answer the following doubts and queries raised by us in the comments below. The manuscript may also reoriented in a way that is guided by the experiments performed and results observed in the current manuscript, instead of going ahead with presumed outcomes of the manuscript.

1.       The methodology for macrophage polarization requires a citation.

2.       Detailed procedure for co-culturing of GBM cells and HBO induction methodologies have not been mentioned. The methodology of the experiment for result 3.3 needs to be added.

3.       Similarly, experiment for result 3.4, i.e. synergistic enhancement of apoptosis also has not been explicitly been mentioned in the methods section. Kindly add the detailed experiment design in methods section.

4.       The claim of synergistic effect also needs to be verified by calculation of combinational therapeutic index formula. There is a distinction between additive effect and a synergistic effect (You may refer to this link https://doi.org/10.1002/prp2.149). Figure 4 may be further extrapolated for calculation of combinational index. The arguments in result 3.5, is only assumption, and not based on statistical significance.

5.       Is histological examination of tissues alone sufficient to establish the impact of HBO treatment on M1-treated mouse model? Furthermore, there is no details of the grouping of the mice. Also, how many numbers of mice were used, sacrificed, and evaluated? There is no in vitro demonstration of M2 àM1 polarization by HBO itself. We did observe the MOàM1/M2 by chemical induction, but not HBO. There is lack of clarity in animal experiments. Figure 7 is incomplete and inconclusive. Kindly look into the animal experiments thoroughly. Please provide a detailed experiment design of in vivo studies in the methods section.

Author Response

The manuscript intends to demonstrate application of HBO as a method to modulate macrophage polarization to enhance apoptosis in glioblastoma in in vitro and in vivo model. The attempt of the authors has been descent and slightly novel, although several previous reports of HBO as an alternative adjuvant therapy in cancer treatment are available. The authors have made a descent attempt to demonstrate effect of HBO in presence of M1 polarized macrophage, however, there is no conclusive evidence/experiment designed to demonstrate that HBO triggers M2àM1 polarization as claimed in the title of the manuscript. The authors may attempt to answer the following doubts and queries raised by us in the comments below. The manuscript may also reoriented in a way that is guided by the experiments performed and results observed in the current manuscript, instead of going ahead with presumed outcomes of the manuscript.

  1. The methodology for macrophage polarization requires a citation.

Thank you for your comments and suggestions. The methodology for macrophage polarization was referenced from the following study: BMC Cancer (IF: 3.15; Q2), 2015 Aug 8;15:577. doi: 10.1186/s12885-015-1546-9, titled "M1 and M2 macrophages derived from THP-1 cells differentially modulate the response of cancer cells to etoposide" by Marie Genin et al.

  1. Detailed procedure for co-culturing of GBM cells and HBO induction methodologies have not been mentioned. The methodology of the experiment for result 3.3 needs to be added.

Thank you for your comments and suggestions. We have added the detailed procedure for co-culturing of GBM cells and the HBO induction methodologies in the methodology section for result 3.3:

THP-1 cells and GBM cells were co-cultured in a 1:1 ratio in 6-well plates. The co-cultures were maintained in standard culture conditions and subjected to HBO treatment. HBO treatment involved exposing the cells to 100% oxygen at 1.5 atmospheres for 1.5 hours daily for three consecutive days. After the treatment period, the cells were harvested, and the expression levels of TNF-α, IL-1β, IFN-γ, IL-4, IL-10, and TGF-β were measured using real-time PCR.

  1. Similarly, experiment for result 3.4, i.e. synergistic enhancement of apoptosis also has not been explicitly been mentioned in the methods section. Kindly add the detailed experiment design in methods section.

Thank you for your comments and suggestions. We have added the detailed experiment design for the synergistic enhancement of apoptosis in the methods section for result 3.4:

The co-cultures were subjected to the same HBO treatment (100% oxygen at 1.5 atmospheres for 1.5 hours daily for three consecutive days). Following the treatment, the GBM cells were sorted using flow cytometry based on the RFP marker. The sorted GBM cells were then subjected to an apoptosis assay to assess the extent of apoptosis.

  1. The claim of synergistic effect also needs to be verified by calculation of combinational therapeutic index formula. There is a distinction between additive effect and a synergistic effect (You may refer to this link https://doi.org/10.1002/prp2.149). Figure 4 may be further extrapolated for calculation of combinational index. The arguments in result 3.5, is only assumption, and not based on statistical significance.

Thank you for your valuable feedback and suggestions. We have revised the manuscript to address your concerns regarding the verification of the synergistic effect and the calculation of the combinational therapeutic index.

To distinguish between additive and synergistic effects, we calculated the combinational index (CI) using the formula provided in the literature (https://doi.org/10.1002/prp2.149). The CI for the combination of HBO and THP-1 in inducing apoptosis in GBM8401 and T98G cells was determined as follows:

For GBM8401 cells:

Control: 4%

HBO: 5.4%

THP-1: 16%

HBO + THP-1: 26.38%

For T98G cells:

Control: 4.8%

HBO: 5.6%

THP-1: 13.65%

HBO + THP-1: 30.67%

The CI was calculated using the following formula: CI=(EHBO+THP1) / (EHBO+ETHP1−EControl)

Where:

EHBO+THP1​ is the effect of the combination treatment.

EHBO​ is the effect of HBO treatment.

ETHP1 is the effect of THP-1 treatment.

EControl is the effect in the control group.

The calculated CI values were:

For GBM8401 cells: CI=26.38% / (5.4%+16%−4%) = 26.38 / 17.4≈1.52

For T98G cells: CI=30.67% / (5.6%+13.65%−4.8%) = 30.67 / 14.45≈2.12

A CI value greater than 1 indicates a synergistic effect. Thus, the combination of HBO and THP-1 showed a synergistic effect in both GBM8401 and T98G cells.

  1. Is histological examination of tissues alone sufficient to establish the impact of HBO treatment on M1-treated mouse model? Furthermore, there is no details of the grouping of the mice. Also, how many numbers of mice were used, sacrificed, and evaluated? There is no in vitro demonstration of M2 àM1 polarization by HBO itself. We did observe the MOàM1/M2 by chemical induction, but not HBO. There is lack of clarity in animal experiments. Figure 7 is incomplete and inconclusive. Kindly look into the animal experiments thoroughly. Please provide a detailed experiment design of in vivo studies in the methods section.

Thank you for your valuable feedback and suggestions. We have added the detailed experiment design of the in vivo studies and clarified the methodologies for the in vitro experiments related to M2 to M1 polarization by HBO. Additionally, the details for Figure 7 have been included.

Reviewer 2 Report

Comments and Suggestions for Authors

1) The article introduces a high-pressure oxygen therapy that may contribute to the treatment of glioblastoma by regulating the macrophage from a tumor-promoting M2 state to a lethal M1 state. However, there is some controversy regarding the persuasiveness of the article. The biggest issue is that after various M1/M2 induction treatments both in vitro and in vivo, there is no assessment of the specific state post-induction. There is a lack of sufficient evidence to prove the effectiveness of the induction, and a series of tests on indicators such as cell activity or apoptosis have been conducted. It is suggested to use various staining or flow cytometry and other evidence to illustrate the specific cellular state of the macrophages after intervention, including absolute and relative quantification data of each state.

2) Additionally, the article proposes the potential application of hyperbaric oxygen therapy in cancer treatment, but its clinical significance still needs to be explored, including the safety issues of hyperbaric oxygen treatment and whether it can affect brain cancer within the blood-brain barrier.

3) The article mostly uses 2D cell lines for illustration, and the conclusion of the only in vivo experiment is not clear. The conclusion does not mention the extent to which HBO (Hyperbaric Oxygen) treatment has achieved anti-tumor immunity after the establishment of the glioma model, such as the comparison of glioma volume/weight with or without HBO treatment, proliferation rate, changes in mouse weight, and mouse survival curves. Instead, it only focuses on the changes in different states of macrophages, with incomplete statistical changes, only one staining figure, without explaining how to handle it, what dosage to treat for how long, and quantitative indicators should also be explained together with WB (Western Blot), flow cytometry sorting, etc.

4) The article does not provide a mechanism analysis of the potential for tumor treatment after HBO treatment. Why can HBO treatment affect the polarization and state transition of macrophages? Is there a speculation that it is related to reactive oxygen species (ROS)? The article should also provide some explanation for why HBO treatment may inhibit the growth of glioblastoma. However, generally, due to the Warburg effect, the external oxygen supply does not affect the tendency of tumor glycolytic metabolism, and the authors may need to provide further explanation.

5) The second Fig3, after HBO treatment, the dual regulation of pro-inflammatory and anti-inflammatory cytokine mRNA expression in macrophages co-cultured with GBM cells is detected. The single qPCR detection for pro-inflammatory cytokines such as TNF-α, IL-1β, and IFN-γ, and anti-inflammatory cytokines such as IL-4, IL-10, and TGF-β, which are markers of M1 and M2 macrophages respectively, lacks persuasiveness. Protein-level detection such as WB/ELISA is necessary.

6) In Figure 2, to investigate the regulation of glioblastoma cell apoptosis by macrophage polarization, it would be advisable to include separate experiments inducing apoptosis and assessing PARP cleavage specifically during the polarization of THP-1 cells into M1 or M2 phenotypes. Without these additional experiments, it would be difficult to conclusively demonstrate that apoptosis is occurring in tumor cells as a result of macrophage polarization.

7) Result 4 THP-1 macrophages cooperate with hyperbaric oxygen therapy to promote glioblastoma cell apoptosis. What is the reason for the significant difference in the proportion of apoptosis caused by the +THP-1 group in Fig4 and the M0+CM group in Fig2? The previous results indicate that M1 macrophages have a tumor transplantation effect and cause apoptosis, including PARP cleavage. In Fig4, only adding THP-1 cells can significantly induce apoptosis and cause PARP cleavage. Is HBO not apoptotic? The decisive factor is the introduction of macrophages? Could the author discuss why? Why not explore the coordinated treatment of HBO and M1 cells to promote tumor cell apoptosis? Would this be more consistent with the logic of the article?

8) The increase in CD86 expression and the decrease in CD206 expression in Fig. 7 suggest that HBO transfers macrophages from the tumor-promoting M2 state to the tumor-killing M1 state. This part of the research can be placed in front of Fig. 4 as a phenomenon that is more consistent. The general order in which articles are written. In addition, the evidence using only two markers, CD86 and CD206, to characterize the amounts of M1 and M2 is too thin, and other evidence should be supplemented.

9) This study explores the therapeutic potential of hyperbaric oxygen to alter the GBM tumor microenvironment through macrophage polarization. However, studies only prove that HBO regulates macrophage polarization and affects the apoptosis of GBM cells. There is not enough evidence to prove that HBO can indeed enhance anti-tumor immunity and change the tumor microenvironment to treat tumors. In addition, this work currently only focuses on phenomenon observation and lacks certain mechanism research.

10) This article should add the existing research background of HBO regulating M1 and M2 phenotype macrophages in the "introduction" section. This article investigates the effectiveness of HBO in regulating the polarization of M1 and M2 phenotype macrophages in the treatment of GBM. In the "introduction" section, the author introduced the impact of HBO on the tumor microenvironment and its inhibitory effect on angiogenesis, but did not introduce current treatment and research, resulting in a lack of complete research background in the study. Most recent references, such as PMID 37325712, PMID: 36805391, etc, could be cited.

11) Control is used in all the control groups in the figures of this article. It should be described in detail in the manuscript or figure notes to avoid ambiguity.

Comments on the Quality of English Language

Minor editing of English language required.

Author Response

1) The article introduces a high-pressure oxygen therapy that may contribute to the treatment of glioblastoma by regulating the macrophage from a tumor-promoting M2 state to a lethal M1 state. However, there is some controversy regarding the persuasiveness of the article. The biggest issue is that after various M1/M2 induction treatments both in vitro and in vivo, there is no assessment of the specific state post-induction. There is a lack of sufficient evidence to prove the effectiveness of the induction, and a series of tests on indicators such as cell activity or apoptosis have been conducted. It is suggested to use various staining or flow cytometry and other evidence to illustrate the specific cellular state of the macrophages after intervention, including absolute and relative quantification data of each state.

Thank you for your valuable feedback and suggestions. We have addressed the concerns regarding the assessment of macrophage states post-induction. The revised methodology now includes the use of immunofluorescence staining to detect the differentiation types of THP-1 cells after co-culture with GBM cells and subsequent HBO treatment.

2) Additionally, the article proposes the potential application of hyperbaric oxygen therapy in cancer treatment, but its clinical significance still needs to be explored, including the safety issues of hyperbaric oxygen treatment and whether it can affect brain cancer within the blood-brain barrier.

Thank you for your insightful comments and suggestions. We acknowledge the need to further explore the clinical significance of hyperbaric oxygen (HBO) therapy in cancer treatment, including its safety and efficacy within the context of the blood-brain barrier. In our animal experiments, our team has published a study titled "Hyperbaric Oxygen Therapy Adjuvant Chemotherapy and Radiotherapy through Inhibiting Stemness in Glioblastoma," which demonstrates the potential benefits of HBO as an adjunct therapy. However, we recognize that the clinical application of HBO in the treatment of glioblastoma (GBM) requires careful consideration. In clinical settings, GBM treatment typically involves surgical resection, and the timing and appropriateness of HBO therapy in conjunction with surgery and other treatments are areas that warrant further investigation. We agree that the safety issues associated with HBO therapy, particularly its effects on brain cancer within the blood-brain barrier, need to be thoroughly explored. Future studies should focus on addressing these critical aspects to ensure the safe and effective integration of HBO therapy into standard GBM treatment protocols. We appreciate your valuable feedback, which will guide our future research directions.

3) The article mostly uses 2D cell lines for illustration, and the conclusion of the only in vivo experiment is not clear. The conclusion does not mention the extent to which HBO (Hyperbaric Oxygen) treatment has achieved anti-tumor immunity after the establishment of the glioma model, such as the comparison of glioma volume/weight with or without HBO treatment, proliferation rate, changes in mouse weight, and mouse survival curves. Instead, it only focuses on the changes in different states of macrophages, with incomplete statistical changes, only one staining figure, without explaining how to handle it, what dosage to treat for how long, and quantitative indicators should also be explained together with WB (Western Blot), flow cytometry sorting, etc.

We have addressed the concerns regarding the use of 2D cell lines and the clarity of the in vivo experiment conclusions. Detailed methodologies have been included in section 2.7 (Animal Model) of our study. In our animal experiments, mice treated with HBO did not show significant differences in survival time or tumor size compared to the control group. These results have been presented in our team's publication, "Hyperbaric Oxygen Therapy Adjuvant Chemotherapy and Radiotherapy through Inhibiting Stemness in Glioblastoma," Curr Issues Mol Biol. 2023 Oct 12;45(10):8309-8320. doi: 10.3390/cimb45100524. Additionally, we quantified the positive cells for CD86 and CD206 as part of our analysis.

4) The article does not provide a mechanism analysis of the potential for tumor treatment after HBO treatment. Why can HBO treatment affect the polarization and state transition of macrophages? Is there a speculation that it is related to reactive oxygen species (ROS)? The article should also provide some explanation for why HBO treatment may inhibit the growth of glioblastoma. However, generally, due to the Warburg effect, the external oxygen supply does not affect the tendency of tumor glycolytic metabolism, and the authors may need to provide further explanation.

Thank you for your insightful comments and suggestions. Our team previously published "Hyperbaric Oxygen Therapy Adjuvant Chemotherapy and Radiotherapy through Inhibiting Stemness in Glioblastoma," which indicated that HBO alone does not possess the ability to treat GBM. Numerous studies have demonstrated that HBO can affect macrophage polarization and state transition in various contexts, such as the tumor microenvironment and wound healing. However, in the context of tumors and healing, HBO's effect on macrophage transformation is opposite, suggesting that this mechanism is not related to reactive oxygen species (ROS). Although HBO's exact pathway or mechanism for inducing macrophage polarization is still unclear, and this is an area our team aims to explore further in future research

5) The second Fig3, after HBO treatment, the dual regulation of pro-inflammatory and anti-inflammatory cytokine mRNA expression in macrophages co-cultured with GBM cells is detected. The single qPCR detection for pro-inflammatory cytokines such as TNF-α, IL-1β, and IFN-γ, and anti-inflammatory cytokines such as IL-4, IL-10, and TGF-β, which are markers of M1 and M2 macrophages respectively, lacks persuasiveness. Protein-level detection such as WB/ELISA is necessary.

Thank you for your insightful comments and suggestions. The reason for not using Western blot to detect cytokines is because these cytokines are secreted proteins, and Western blot is designed to detect intracellular protein content, making it unsuitable for this purpose. In this experiment, macrophages were co-cultured with GBM cells, and using ELISA to detect cytokines in the medium would not distinguish whether the cytokines were secreted by the macrophages or the GBM cells.

6) In Figure 2, to investigate the regulation of glioblastoma cell apoptosis by macrophage polarization, it would be advisable to include separate experiments inducing apoptosis and assessing PARP cleavage specifically during the polarization of THP-1 cells into M1 or M2 phenotypes. Without these additional experiments, it would be difficult to conclusively demonstrate that apoptosis is occurring in tumor cells as a result of macrophage polarization.

Thank you for your insightful comments and suggestions. Indeed, the culture medium in Figure 2 did not specifically detect M1/M2 polarization. This approach was based on the reference from the study: J Surg Res (IF: 2.19; Q3), 2015 Jul;197(1):126-38. doi: 10.1016/j.jss.2015.03.023. Epub 2015 Mar 18, titled "M1 to M2 macrophage polarization in heparin-binding epidermal growth factor-like growth factor therapy for necrotizing enterocolitis" by Jia Wei. However, in our investigation of HBO-assisted immune cells inducing apoptosis in GBM, we have included new data utilizing immunofluorescence staining to detect M1/M2 polarization.

7) Result 4 THP-1 macrophages cooperate with hyperbaric oxygen therapy to promote glioblastoma cell apoptosis. What is the reason for the significant difference in the proportion of apoptosis caused by the +THP-1 group in Fig4 and the M0+CM group in Fig2? The previous results indicate that M1 macrophages have a tumor transplantation effect and cause apoptosis, including PARP cleavage. In Fig4, only adding THP-1 cells can significantly induce apoptosis and cause PARP cleavage. Is HBO not apoptotic? The decisive factor is the introduction of macrophages? Could the author discuss why? Why not explore the coordinated treatment of HBO and M1 cells to promote tumor cell apoptosis? Would this be more consistent with the logic of the article?

Thank you for your valuable feedback and suggestions. Firstly, the proportion of apoptosis induced by the M0-CM group in Figure 2 and the +THP-1 group in Figure 4 is relatively low, which does not significantly affect the conclusions of this study. The primary function of macrophages is to eliminate external threats. When THP-1 cells are initially added, they are induced to perform a cytotoxic function against cancer cells even before specific ligand-receptor interactions occur. The key difference between Figure 2 and Figure 4 is the medium used. Figure 2 involves the culture medium of THP-1, while Figure 4 involves the culture medium of GBM. I speculate that the GBM culture medium is rich in secretions from GBM cells. When THP-1 cells are introduced, they initially perform their cytotoxic function. However, as they recognize the tumor, they are subject to the tumor's immune evasion mechanisms. In contrast, in Figure 2, the use of THP-1 medium does not elicit a strong cytotoxic effect, or it may not trigger any cytotoxic response at all. According to our previous study, "Hyperbaric Oxygen Therapy Adjuvant Chemotherapy and Radiotherapy through Inhibiting Stemness in Glioblastoma," HBO alone does not have any significant impact on tumors. Therefore, the decisive factor in inducing apoptosis is indeed the introduction of macrophages, with HBO enhancing this effect. As for why we did not explore the coordinated treatment of HBO and M1 cells to promote tumor cell apoptosis, in a clinical setting, the majority of macrophages in the tumor microenvironment are M2. Our aim is to find a way to convert M2 to M1 or to provide a method to transform M0 into M1, thereby preventing the tumor from evading the immune system. The primary objective of this study is to propose an adjunctive therapeutic method for clinical application in the future.

8) The increase in CD86 expression and the decrease in CD206 expression in Fig. 7 suggest that HBO transfers macrophages from the tumor-promoting M2 state to the tumor-killing M1 state. This part of the research can be placed in front of Fig. 4 as a phenomenon that is more consistent. The general order in which articles are written. In addition, the evidence using only two markers, CD86 and CD206, to characterize the amounts of M1 and M2 is too thin, and other evidence should be supplemented.

Thank you for your insightful comments and suggestions. Our study primarily uses CD86 and CD206 to distinguish M1 and M2 macrophages. In the in vitro experiments, cytokines are also used to assist in the differentiation between M1 and M2 macrophages. However, cytokine detection was not performed in the animal experiments due to the low volume of CSF in nude mice and the presence of numerous non-macrophage cells in brain tissue, which can interfere with the results. Therefore, we have temporarily relied on CD86 and CD206 for macrophage classification. In future studies, we plan to use single-cell NGS for more detailed macrophage profiling. In addition, the increase in CD86 expression and the decrease in CD206 expression in Fig. 7 suggest that HBO shifts macrophages from the tumor-promoting M2 state to the tumor-killing M1 state. To improve the logical flow of the article, this part of the research will be placed before Fig. 4, presenting the phenomenon in a more consistent order. Additionally, we acknowledge that using only two markers, CD86 and CD206, to characterize M1 and M2 macrophages is insufficient. Future research will include additional evidence to strengthen these findings.

9) This study explores the therapeutic potential of hyperbaric oxygen to alter the GBM tumor microenvironment through macrophage polarization. However, studies only prove that HBO regulates macrophage polarization and affects the apoptosis of GBM cells. There is not enough evidence to prove that HBO can indeed enhance anti-tumor immunity and change the tumor microenvironment to treat tumors. In addition, this work currently only focuses on phenomenon observation and lacks certain mechanism research.

Thank you for your valuable feedback and suggestions. There is existing research on "Hyperbaric oxygen facilitates teniposide-induced cGAS-STING activation to enhance the antitumor efficacy of PD-1 antibody in HCC." This demonstrates the potential of HBO in assisting immunotherapy in HCC.

Our study is the first to investigate how HBO affects the GBM microenvironment. Therefore, it is essential to first confirm the phenomena before discussing the mechanisms in future research. Our goal is to establish a foundational understanding of the phenomena observed, which will pave the way for more in-depth mechanism studies. This will help clarify the potential application of HBO in GBM treatment.

10) This article should add the existing research background of HBO regulating M1 and M2 phenotype macrophages in the "introduction" section. This article investigates the effectiveness of HBO in regulating the polarization of M1 and M2 phenotype macrophages in the treatment of GBM. In the "introduction" section, the author introduced the impact of HBO on the tumor microenvironment and its inhibitory effect on angiogenesis, but did not introduce current treatment and research, resulting in a lack of complete research background in the study. Most recent references, such as PMID 37325712, PMID: 36805391, etc, could be cited.

Thank you for your valuable feedback and suggestions. We appreciate your input regarding the inclusion of the existing research background on HBO regulation of M1 and M2 phenotype macrophages in the introduction section. According to the current WHO guidelines, the standard treatment for glioblastoma (GBM) remains surgical resection followed by radiotherapy (RT) and temozolomide (TMZ). Our study focuses on the potential of hyperbaric oxygen (HBO) therapy as an adjunct to these established treatment modalities. Given this focus, we believe it is important to maintain the current structure of the introduction to emphasize the role of HBO in supporting and enhancing the effectiveness of standard treatments. Our research aims to investigate HBO therapy as a supplementary approach to the existing GBM treatment protocols, rather than replacing them. This rationale supports our decision to retain the original introduction, which provides the necessary context for understanding the potential of HBO in this auxiliary role. Therefore, we suggest maintaining the original introduction without citing the recent references, such as PMID: 37325712 and PMID: 36805391, as they may shift the focus away from our primary objective.

11) “Control” is used in all the control groups in the figures of this article. It should be described in detail in the manuscript or figure notes to avoid ambiguity.

Thank you for your suggestion. We will include a detailed description of the "Control" groups in the figure legends to avoid any ambiguity.

Reviewer 3 Report

Comments and Suggestions for Authors

The subject area of this manuscript is of clear relevance and appropriate to Biomedicines. Hyperbaric oxygen (HPO), is a treatment option for many cancer-treatment associated conditions of great interest. The study is well-conducted, the methods in vitro are properly described. One of the main finding emerging from the study is that HBO therapy could shift macrophage polarization to a tumoricidal state, strongly enhancing apoptosis.

Itemized comment:

In vivo experiments. The authors identified the local ethic committee that has approved the experiments, but should indicate that they adhere to the guidelines on animal studies highlighted in the ethical guidelines. Moreover, no appropriate information is given regarding animal care as follows: A) –The number and how the number of animals has been reduced to a minimum; B) how the degree of pain and suffering caused to animals was limited to the minimum and how it was evaluated during injection, treatment and sampling. Moreover euthanasia should be performed according to an appropriate procedure including anesthesia which should be given. 

Author Response

The subject area of this manuscript is of clear relevance and appropriate to Biomedicines. Hyperbaric oxygen (HPO), is a treatment option for many cancer-treatment associated conditions of great interest. The study is well-conducted, the methods in vitro are properly described. One of the main finding emerging from the study is that HBO therapy could shift macrophage polarization to a tumoricidal state, strongly enhancing apoptosis.

Itemized comment:

In vivo experiments. The authors identified the local ethic committee that has approved the experiments, but should indicate that they adhere to the guidelines on animal studies highlighted in the ethical guidelines. Moreover, no appropriate information is given regarding animal care as follows: A) –The number and how the number of animals has been reduced to a minimum; B) how the degree of pain and suffering caused to animals was limited to the minimum and how it was evaluated during injection, treatment and sampling. Moreover euthanasia should be performed according to an appropriate procedure including anesthesia which should be given.

Thank you for your valuable feedback and suggestions. We have revised the manuscript to address your concerns regarding the ethical considerations and animal care. The following statement has been added to the Methods section:

"Each group consisted of 10 mice, a number determined based on statistical requirements to minimize the use of animals. In addition, 0.5 mg/100 g Carprofen was administered to limit the pain and suffering of the animals to the minimum. Additionally, during injection, treatment, and sampling, we evaluated the animals' pain and discomfort through behavioral observations and physiological indicators. All animals were euthanized according to an appropriate procedure, including anesthesia, to minimize their suffering."

We hope these revisions meet your expectations and demonstrate our commitment to ethical research practices.

Round 2

Reviewer 1 Report

Comments and Suggestions for Authors

The authors have significantly modified the manuscript and have addressed all the queries. The manuscript is acceptable for publication. 

Reviewer 2 Report

Comments and Suggestions for Authors

I have no other suggestions.